# Risk factors analysis for neglected human rickettsioses in rural communities in Nan province, Thailand: A community-based observational study along a landscape gradient

**Kittipong Chaisiri**[1]*, **Ampai Tanganuchitcharnchai**[2], **Anamika Kritiyakan**[3], **Chuanphot Thinphovong**[3], **Malee Tanita**[4], **Serge Morand**[1,3,5], **Stuart D. Blacksell**[2,6]*

1 Department of Helminthology, Faculty of Tropical Medicine, Mahidol University, Bangkok, Thailand, 2 Mahidol-Oxford Tropical Research Medicine Unit (MORU), Faculty of Tropical Medicine, Mahidol University, Bangkok, Thailand, 3 Faculty of Veterinary Technology, Kasetsart University, Bangkok, Thailand, 4 Saen Thong Health Promoting Hospital, Tha Wang Pha, Nan, Thailand, 5 Faculty of Veterinary Technology, CNRS ISEM–CIRAD ASTRE, Kasetsart University, Bangkok, Thailand, 6 Center for Tropical Medicine & Global Health, Nuffield Department of Medicine, Oxford University, Oxford, United Kingdom

* kittipong.cha@mahidol.ac.th (KC); stuart.blacksell@ndm.ox.ac.uk (SDB)

## Abstract

In this study, we estimated exposure for Scrub typhus (STG), Typhus (TG) and Spotted fever groups (SFG) rickettsia using serology at a fine scale (a whole sub-district administration level) of local communities in Nan Province, Thailand. Geographical characteristics of the sub-district were divided into two landscape types: lowland agricultural area in an urbanized setting (lowland-urbanized area) and upland agricultural area located close to a protected area of National Park (upland-forested area). This provided an ideal contrast between the two landscapes with low and high levels of human-altered habitats to study in differences in disease ecology. In total, 824 serum samples of participants residing in the eight villages were tested by screening IgG ELISA, and subsequently confirmed by the gold standard IgG Immunofluorescent Assay (IFA). STG and TG IgG positivity were highest with seroprevalence of 9.8% and 9.0%, respectively; whereas SFG positivity was lower at 6.9%. Inhabitants from the villages located in upland-forested area demonstrated significantly higher STG exposure, compared to those villages in the lowland-urbanized area (chi-square = 51.97, p < 0.0001). In contrast, TG exposure was significantly higher in those villagers living in lowland-urbanized area (chi-square = 28.26, p < 0.0001). In addition to the effect of landscape types, generalized linear model (GLM) analysis identified socio-demographic parameters, i.e., gender, occupation, age, educational level, domestic animal ownership (dog, cattle and poultry) as influential factors to explain the level of rickettsial exposure (antibody titers) in the communities. Our findings raise the public health awareness of rickettsiosis as a cause of undiagnosed febrile illness in the communities.

**Data Availability Statement:** All relevant data are within the manuscript and its Supporting Information files.

**Funding:** SM received the French ANR Project FutureHealthSEA (grant number: ANR-17-CE35-0003-02) "Predictive scenarios of health in Southeast Asia: linking land use and climate changes to infectious diseases". SDB is funded by the Wellcome Trust of the United Kingdom [220211]. This research was funded in whole, or in part, by the Wellcome Trust [220211]. The funders had no role in study design, data collection and analysis, decision to publish, or preparation of the manuscript.

**Competing interests:** The authors have declared that no competing interests exist.

## Author summary

Evidence of human exposures to rickettsial pathogens were reported from a cross-sectional study at a whole sub-district scale of local communities in Nan Province, Thailand. Seroprevalence and level of rickettsial exposures demonstrated differences between the habitat types, ecological aspects and socio-demographic factors. In addition, abundance of domestic animals in the community appeared to be one of significant factors influencing levels of human exposure to rickettsial pathogens. Our findings will benefit the local public health by raising awareness of rickettsial infections as one of potential health concerns in the community. Inclusion of rickettsioses in routine laboratory diagnosis would help to differentiate unknown febrile illness and guide appropriate treatment. Further studies are required, particularly in the fields of disease ecology as well as medical and veterinary entomology, in order to better understand epidemiology and potential zoonotic transmission of these neglected rickettsioses in endemic areas.

## Introduction

Human rickettsioses are neglected emerging and re-emerging diseases caused by a group of obligate intracellular bacteria in the Order Rickettsiales (e.g., *Rickettsia* and *Orientia*). Rickettsial pathogens causing disease in humans can be divided into the three main groups, known as scrub typhus (STG), typhus (TG) and spotted fever groups (SFG) [1]. These zoonotic bacteria are transmitted to humans and animals after bites by arthropod vectors, including ticks, mites, fleas or lice. Co-occurrences of rickettsial species among arthropod vectors, domestic animals and humans as well as increasing opportunities for interactions at the human-animal interface are of importance in zoonotic transmission cycles [2–4]. These vector-borne diseases have increasingly been recognized as one of the common causes of febrile illness in Southeast Asia, in addition to dengue, leptospirosis and malaria [1,5,6].

Scrub typhus is a chigger-borne rickettsiosis caused by intracellular bacteria in the genus *Orientia*, including *Orientia tsutsugamushi* (reported primarily in Asia-Pacific region), *Orientia chuto* (in the Arabian Peninsula and African countries) *and Orientia sp. (Candidatus Orientia chiloensis*, a potential new species from Chile) [7–11]. Larval stage of mites in the family Trombiculidae, known as "chiggers", are the main vector of the disease. Previous studies have suggested that human scrub typhus is more common in rural or sub-urban areas [12,13] with the majority of disease in farmers, military soldiers and jungle trekkers [7,14,15]. Typhus group (TG) mainly involves murine typhus (also known as endemic typhus) infections caused by *Rickettsia typhi*, a flea-borne rickettsiosis with worldwide distribution, notably in tropical coastal regions [16,17]; and *Rickettsia prowazekii* which causes epidemic typhus, a louse-borne rickettsiosis in the areas of poor socioeconomic status and high prevalence of human body louse infestation [18]. Transmission of typhus group rickettsioses generally occur when the bacteria in the feces of arthropod vectors are rubbed into the bite wound or the vectors themselves are crushed and broke in to the skin [19]. For the spotted fever group (SFG), there are more than 25 rickettsial species and related strains distributed globally, and known to cause disease in human [20], including *Rickettsia africae, Rickettsia asiatica, Rickettsia australis, Rickettsia conorii, Rickettsia heilongjiangensis, Rickettsia helvetica, Rickettsia honei, Rickettsia japonica, Rickettsia parkeri, Rickettsia raoultii, Rickettsia rickettsia, Rickettsia sibirica* and *Rickettsia tamurae*. These SFG rickettsia are mainly harbored by ticks, and transmitted to human or other animal hosts via their bites. Other ectoparasites such as fleas and mites are also involved as the vectors of several SFG rickettsia including the flea-borne *Rickettsia felis* and the mite-borne *Rickettsia akari* [2,21,22].

Diagnosis of these pathogens relies on molecular techniques such as PCR-based amplification to demonstrate the presence of the infectious agent, as well as serological methods as a surrogate including enzyme-linked immunosorbent assay (ELISA) and immunofluorescence assays (IFA) [23]. Serological investigations allow epidemiologists to investigate the history of rickettsial exposure and estimation of seroprevalence by detecting the occurrence of preexisting levels of antibodies in a population [24–26]. IFA detection of IgM and/or IgG is recognized as a gold standard for rickettsial serological diagnosis, particularly for STG, however there remains uncertainty as to the most suitable diagnostic cut-off for each geographic location [23,27]. ELISA-based diagnostics give acceptable levels of sensitivity and specificity for STG, SFG and TG antibody detection [23, 28,29] as well as having increased diagnostic throughput and less subjectivity [29,30]. However, serological results are normally confined to the serogroup level. Although generally, there are cross-reactions within the serogroups, they occur less so between serogroups with the exception that cross-reactions between SFG and TG antibodies are recognised to a limited extent [1,31].

In Thailand, previous studies have indicated that all three rickettsial groups exist and are a cause for public health concern [2,5,32–34]. In terms of human febrile illness, scrub typhus is the most prevalent rickettsial pathogen which has markedly increased during the last two decades although this may be due to increased awareness and access to improved diagnostics. During 2003–2018, there were approximately 100,000 cases reported to the National Disease Surveillance system (Ministry of Public Health, Thailand). The disease is reported nationwide, with the majority of the cases from the northern region accounting approximately half (53%) of the total reported cases per year [34]. Murine typhus is also recognized a rickettsiosis in Thailand. Case reports of *R. typhi* infection have been documented in several provinces, either in municipalities, rural communities or border areas [5,35–38]. Epidemiological data regarding SFG rickettiosis is rather more scarce compared to scrub typhus and murine typhus. In 1994, three human SFG rickettsiosis cases from Chiang Mai Province were confirmed by serology, and documented as the first report in Thailand [39]. The first human case of *R. felis* infection in Thailand (and in Asia) was reported at the Thai-Myanmar border in Kanchanaburi Province (32). Thai tick typhus *Rickettsia* TT-118 (also known as *R. honei*) was identified from a patient in urban area of Bangkok who had history of camping at a National Park prior to the onset of fever [40]. In addition, *Rickettsia* spp. closely related to *R. japonica* and *R. helvetica* have also reported to infect humans in Thailand [41–43].

Most of the case reports and epidemiological studies in Thailand were recognised in the context of passive surveillance or retrospective studies from stored clinical samples [44], while active surveillance data of human rickettsiosis is still limited. Here, within the research framework of FutureHealthSEA project (Predictive scenarios of health in Southeast Asia: linking land use and climate changes to infectious diseases, ANR-17-CE35-0003), we performed screening for human rickettsiosis at the sub-district administration level of local communities in Nan Province, northern Thailand. The objectives of the study were: (1) to estimate level of past exposure to rickettsial pathogens using serological diagnosis approach; and (2) to determine potential factors, such as socio-demographic, zoonotic potential and environmental parameters influencing rickettsial exposures in a local scale setting.

## Materials and methods

### Ethics statement

All procedures involving human samples and data collection were reviewed and approved by the Ethics Committee of the Faculty of Tropical Medicine, Mahidol University (document no. MUTM 2018-035-01). Signed formal consent form was obtained from all the participants and also from the parent/guardian of the child participants.

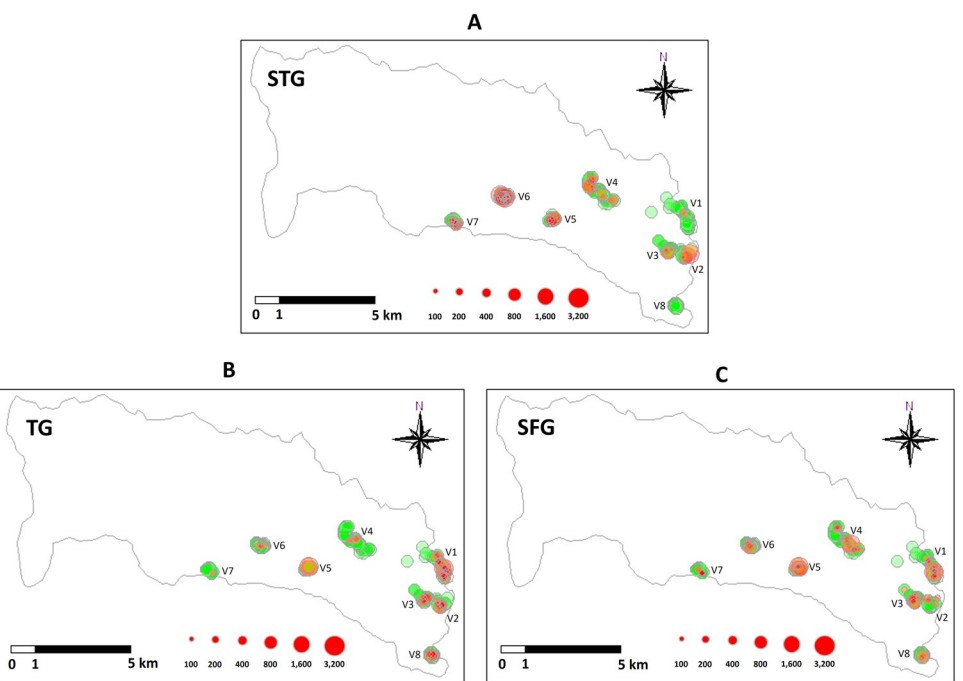

**Fig 1. Distribution maps demonstrating the level of rickettsiosis exposures in participants from the eight villages in Saen Thong Subdistrict, Nan Province.** Distribution and level of (A) STG, (B) TG and (C) SFG individual exposures from each village. Green circles indicate negativity. Red circles show positive exposures, and size of the circles indicate level of IgG IFA positivity at a diagnostic titer cut-off of 1:100.

## Study population and individual demographic information

In 2018, we conducted a cross-sectional study at the local communities, all of the 8 villages in Saen Thong Sub-district, Tha Wang Pha District, Nan Province (latitude 19.133 and longitude 100.7680). Geographical distribution of the 8 villages is shown in Fig 1. In terms of geographical characteristics, the sub-district can be roughly divided into the two landscape types: (1) lowland agricultural area in an urbanized setting (Lowland-urbanized area), closed to Tha Wang Pha city on the east; and (2) upland agricultural area closed to the forested area of Nantaburi National Park (Upland-forested area) on the west.

The target population of this study was randomly selected with the following inclusion criteria: both males and females; age over 9 years old; recently and permanently residing in the study area; able to adequately respond to the questions and voluntary participation in the research project. Individual data regarding participants' personal information was collected including age group (children at 9–17 years old; young adult at 18–35 years old; adult at 36–60 years old; and elderly at >60 years old), gender (male and female) and occupation (farmer or non-farmer). Information on companion pets and livestock ownership, which included numbers of dogs, cats, poultry, pigs and cattle, were also obtained to determine potential zoonotic association. Every household that the participants' resided were geo-tagged to collect GPS coordinates information for further spatial case mapping.

## Blood sample collection and serological diagnosis of rickettsial past exposure

Staff from the local primary health care unit (PCU) made an appointment with the village chiefs for preparation of participant recruitment for blood collection. Blood samples were

collected via venipuncture (5ml) in non-anticoagulant tubes (VACUETTE TUBE Serum Clot Activator, Greiner Bio-One, Austria), and then centrifuged at 2,795 xg for 5 min within the same day of collection to separate serum samples in sterile 1.5 ml collection tubes (Axygen, US). The serum samples were frozen in -80˚C prior to laboratory investigations.

Due to the large number of samples, sera were screened for the presence of IgG antibodies at 1:100 dilution using STG, TG or SFG IgG ELISA using diagnostic cut off with a high sensitivity of 0.50 nett OD. ELISA-positive samples were titrated in the "gold standard" IFA to determine the endpoint antibody titer. The MORU in-house STG IgG ELISA used specific antigens of *O. tsutsugamushi* Karp, Kato, Gilliam, and TA716 strains to detect scrub typhus IgG antibodies [45]. For the TG ELISA, *R. typhi* Wilmington strain and for SFG ELISA, *R. conorii* and *R. honei* antigens were applied using the same methodology. Consequently, we performed IgG IFA to produce quantitative results, with the full methodology described elsewhere [46]. Briefly, sera were serially 2-fold diluted from 1:100 to 1:25,600 and the endpoint was determined as the highest titer displaying specific fluorescence. Titer >1:100 was considered positive for the presence of IgG antibodies.

## Statistical analysis

Estimation of STG, TG and SFG by village, landscape type and personal attributes were performed using descriptive statistics, i.e., seroprevalence with 95% confidence interval. A Venn diagram was created using "ggplot2" package [47] in R freeware [48] to illustrate number and proportion of participants exposed with STG, TG, SFG and co-exposures. Seropositive cases were geographically mapped to reveal distribution patterns of STG, TG and SFG using "tmap" [49] and "raster" [50] packages in R freeware [48].

Generalized linear model (GLM) was performed using R freeware [48] to determine socio-demographic and environmental factors explaining the level of STG, TG and SFG exposures (titers). Similar initial models were applied to the three rickettsial groups, comprising with the following variables: Landscape type, Gender, Occupation, Age, Educational level, as well as discrete variables for population numbers of dogs, cats, pigs, poultry and cattle. Poisson distribution was selected for model fitting after checking data distribution using "fitdistrplus" package [51]. GLM with Akaike's information criterion corrected for sample size (*AICc*) was performed with negative binomial distribution using "glmulti" package [52]. The best models for each rickettsial group were selected for further discussion based on low *AICc* and high Akaike's weight (*Wr*) scores [53]. To identify the degree of multicollinearity among explicative variables, the variance inflation factor (VIF) was computed using "car" package [54]. In addition, quality of the selected models was assessed with goodness-of-fit and power analysis tests using "sjstats" [55] and "lmSupport" [56] packages, respectively.

## Results and discussion

### Socio-demographic information of Saen Thong subdistrict

In total, 824 sera from the eight villages were subjected to serological testing yielding a participation rate of 19.8% (total registered population of 4,145 in the subdistrict). Of the total population registered in the system, it is likely that a proportion of the population (particularly those working age) did not live in the area because they moved to work in other cities during the study which may have artificially lowered the participation rate. Serum samples were collected from participants who lived in 524 households covering almost a half (48.3%) of the total number of eligible households (1,083 registered households in total). Adults were the dominant age group (477 samples, 57.8%), followed by the elderly (250 samples, 30.3%), children (45 samples, 5.4%), young adults (37 samples, 4.5%), however 15 samples (1.8%) were not

able to assigned an age group category as this information was not recorded during the interview. The majority of the participants were farmers (601 samples, 72.9%), with non-farmers accounting for 21.9% (181 samples) and unknown occupation for 42 samples (5.1%). We were unable to assign their occupation as this information was not recorded during the interview (more details in S1 Table).

Over a half of the participants (483 individuals, 58.6%) kept or raised animals including companion pets and farm livestock at home, whereas 341 individuals (41.3%) responded as having no animals. Among these, poultry (mainly backyard chickens) were the most popular (375 people, 45.5%) followed by 185 people (22.4%) responded to have dogs, 57 people (6.9%) cats, 14 people (1.7%) cattle and 6 people (0.7%) raised pigs at home.

## Rickettsiosis past exposure in Saen Thong subdistrict: Associations of environmental and socio-demographic factors

Rickettsial exposure was noted in the Saen Thong subdistrict with seroprevalence of STG, TG and SFG of 9.8%, 9.0% and 6.9%, respectively (Table 1). Comparing the results of STG seroprevalence presented here to previous publications, the study population in Nan Province

**Table 1. Seroprevalence of human exposure to rickettsial infections, scrub typhus group (STG), typhus group (TG) and spotted fever group (SFG) in rural communities of Nan Province, Thailand.**

| Category | n | Scrub typhus group (STG) | | Typhus group (TG) | | Spotted fever group (SFG) | |
|---|---|---|---|---|---|---|---|
| | | Number positive | Seroprevalence [95% CI] | Number positive | Seroprevalence [95% CI] | Number positive | Seroprevalence [95% CI] |
| **Village** | | | | | | | |
| *Lowland agricultural landscape* | | | | | | | |
| No.1 –Ban Na Noon | 161 | 2 | 1.2 [0.2–4.5] | 39 | 24.2 [17.9–31.6] | 16 | 9.9 [6.1–15.7] |
| No.2 –Ban Na Sai | 113 | 3 | 2.7 [0.7–7.4] | 9 | 8.0 [4.1–14.4] | 4 | 3.5 [1.2–8.7] |
| No.3 –Ban Pho | 98 | 4 | 4.1 [1.4–10.1] | 7 | 7.1 [3.4–14.1] | 6 | 6.1 [2.7–12.6] |
| No.8 –Ban Hae | 71 | 0 | - | 9 | 12.7 [6.5–22.4] | 4 | 5.6 [1.9–13.8] |
| *Upland forested landscape* | | | | | | | |
| No.4 –Ban Huak | 141 | 13 | 9.2 [5.2–15.1] | 3 | 2.1 [0.6–6.2] | 11 | 7.8 [4.1–13.4] |
| No.5 –Ban Nam Krai | 64 | 8 | 12.5 [5.8–23.2] | 1 | 1.6 [0.1–8.3] | 4 | 6.3 [2.2–15.3] |
| No.6 –Ban Huay Muang | 122 | 32 | 26.2 [19.1–34.8] | 5 | 4.1 [1.6–9.3] | 4 | 3.3 [1.1–8.1] |
| No.7 –Ban Santisuk | 54 | 19 | 35.2 [22.9–49.1] | 1 | 1.9 [0.1–9.8] | 8 | 14.8 [6.9–26.7] |
| **Landscape type** | | | | | | | |
| Lowland agricultural area | 443 | 9 | 2.0 [1.0–3.8] | 64 | 14.4 [11.3–18.0] | 30 | 6.8 [4.7–9.6] |
| Upland forested area | 381 | 72 | 18.9 [15.2–23.2] | 10 | 2.6 [1.4–4.8] | 27 | 7.1 [4.8–10.2] |
| **Gender** | | | | | | | |
| Male | 352 | 56 | 15.9 [12.3–20.1] | 31 | 8.8 [6.2–12.3] | 27 | 7.7 [5.2–10.9] |
| Female | 469 | 25 | 5.3 [3.6–7.7] | 43 | 9.2 [6.8–12.1] | 29 | 6.2 [4.3–8.7] |
| **Age group** | | | | | | | |
| Children (9–17) | 45 | 1 | 2.2 [0.1–11.8] | 1 | 2.2 [0.1–11.8] | 1 | 2.2 [0.1–11.8] |
| Young adult (18–35) | 37 | 0 | - | 2 | 5.4 [0.9–18.5] | 4 | 10.8 [3.8–25.4] |
| Adult (36–60) | 477 | 39 | 8.2 [5.9–11.0] | 44 | 9.2 [6.9–12.1] | 33 | 6.9 [4.9–9.6] |
| Elderly (>60) | 250 | 40 | 16.0 [11.8–21.1] | 27 | 10.8 [7.3–15.3] | 18 | 7.2 [4.5–11.1] |
| **Occupation** | | | | | | | |
| Farmer | 601 | 63 | 10.5 [8.2–13.2] | 59 | 9.8 [7.6–12.4] | 39 | 6.5 [4.7–8.8] |
| Non-farmer | 181 | 14 | 7.7 [4.6–12.6] | 13 | 7.2 [4.0–12.1] | 15 | 8.3 [4.8–13.2] |
| **Total** | **824** | **81** | **9.8 [7.9–12.1]** | **74** | **9.0 [7.2–11.1]** | **57** | **6.9 [5.3–8.8]** |

demonstrated a similar rate of exposure. This agreed with the review data of Bonell et al. (2017) that *Orientia tsutsugamushi* infection endemic in several Asian countries (i.e., Bangladesh, Indonesia, Lao PDR, Malaysia, Papua New Guinea and Sri Lanka), with seroprevalence ranging from 9.3%-27.9% [44]. Compared with previous studies in Thailand, the STG seroprevalence in Nan province was slightly higher than a previous reported among the populations in the three previously studied provinces of Thailand (4.2%), i.e., Chiang Rai, Khon Kaen and Nakhon Phanom provinces [33], however it should be noted that the study used a different antibody cut-off titer ($\geq$ 1:128 compared with $\geq$1:100 in this study to detect IgG using IFA) to define positivity. In addition, our result demonstrated much lower STG positivity rate than the population in the western provinces, i.e., 62.5% Ratchaburi, 64.6% Petchaburi and 49.1% Kanchanaburi province [57], however it is important to recognise that a lower diagnostic cut-off value was used for interpretation (IgG using IFA, $\geq$1:50 compared with $\geq$1:100 in our study). In terms of murine typhus, seroprevalence of TG in the present study was slightly higher (9.0%) than the study in Chiang Rai, Khon Kaen and Nakhon Phanom provinces (4.2%), [33]; and similar rate of exposure with a previous study (8.0%) in suburban Bangkok which used the indirect immunoperoxidase assay with a diagnostic cut-off of $\geq$1:50 [58]. For SFG, our result in Nan province demonstrated higher seroprevalence (6.9%) than that from a previous study in Chiangrai, Khon Kaen and Nakhon Phanom provinces (0.8%), [33]. In contrast, another seropositivity study reported much higher SFG exposures in Chiang Rai (33%) and Tak provinces (27.3%), but the study used different assays (SFGR Enzyme Immunoassay IgG Antibody Kit), [59]. Therefore, it is important to consider the serological diagnostic methods and selection of the diagnostic cut-off values when comparing exposure rates, otherwise this has the potential to cause analysis bias and misinterpretation.

There was evidence of multiple positivity with the three rickettsial serogroups in a small proportion of participants. However, no individual demonstrated multiple positivity with all three rickettsial groups. Multiple positivity was occurred in 10 (5.2%), 6 (3.1%) and 3 (1.6%) cases between STG-SFG, TG-SFG and STG-TG, respectively (Fig 2). However, it is recognised that serological cross-reactivity among SFG and TG [60] may occur, which could account for the dual-positivity between these antigenic groups.

GLM analysis revealed that environmental factors, including landscape type and socio-demographic parameters, such as gender, occupation, age, educational level, and number of domestic animals in possession (dog, cattle and poultry) was significantly associated with higher rickettsial antibody levels in the communities (Table 2). Inhabitants from the villages located in upland-forested area (Village 4, 5, 6 and 7) demonstrated significantly higher STG seroprevalence (9.2% to 35.2%) and higher level of STG exposure (titer), compared to those villages (Village 1, 2, 3 and 8) in the lowland-urbanized area (none to 4.1%), (chi-square = 51.97, p < 0.0001), (Figs 1A and 3). Individuals located in upland-forested areas (increased habitat complexity and biodiversity) would have higher rates of exposures to chigger mites resulting in higher STG seroprevalence than those who live in urban environments (7,34). The reverse trend was observed for TG exposure, where the villages in the lowland-urbanized area demonstrated significantly higher seroprevalence (7.1% to 24.2%) and higher TG titer level compared to those villages in the upland-forested area (1.6% to 4.1%), (chi-square = 28.26, p < 0.0001), (Figs 1B and 4). Individuals located in urban environments may have potentially higher rates of exposure to fleas (the vector for murine typhus) parasitizing on rats and companion pets [61,62] resulting in higher TG seroprevalence compared with those individuals that are located in more forested or non-urban environments. These trends are in accordance with the study of scrub typhus and murine typhus cases in Lao PDR [12] and Malaysia [63]. We found no significant effect of landscape type neither on SFG seroprevalence nor antibody titer level of SFG

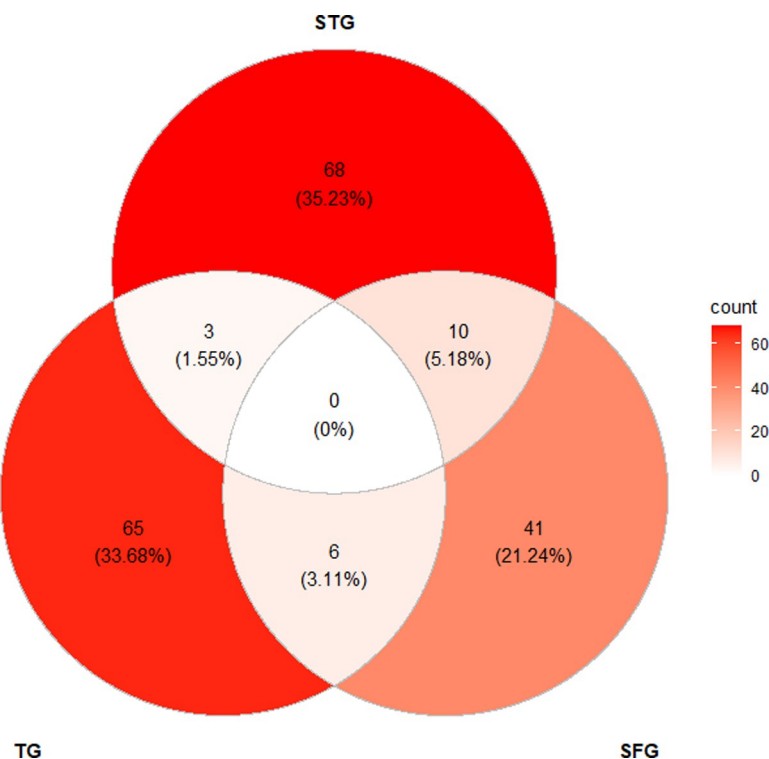

**Fig 2. Venn diagram represents number of participants exposed with STG, TG, SFG and co-exposures.** Relative proportions of the rickettsiosis exposures are provided in parentheses.

exposure. Distribution maps of STG-, TG- and SFG-positive individuals from each village are provided in S1–S3 Figs.

Male participants demonstrated significantly higher STG seroprevalence (15.9%) when compared with females (5.3%), (chi-square = 19.47, p < 0.0001), (see Table 1). This may be explained by the need for males to enter into the forest for hunting or animal tending purposes. For TG and SFG positivity, no clear difference was observed between the two genders. Surprisingly, there was no discernible difference in rickettsial seroprevalence between farmer and non-farmer participants. One would normally associate farming with higher levels of scrub typhus seroprevalence, however it would depend where the farming was performed for example, whether the farming was performed close or remote from forested areas. In terms of educational level, the rickettsial exposures were found to be significantly more common among those with lower levels of education compared to those with higher education which had been previously identified in Lao PDR [12].

### Domestic animals as influencing factor of rickettsiosis exposure in Saen Thong subdistrict

In terms of the human-animal interface, our results demonstrate a significant link between number of domestic animals owned and the level of human rickettsial exposures. Higher levels of STG, SFG and TG exposures were positively correlated with number of dogs, poultry and both dogs and cattle, respectively (Figs 3–5). Domestic animals can act as natural vertebrate reservoir hosts for *Rickettsia* spp. in Southeast Asia [2], together with the close contact between people, domestic animals and associated ectoparasites (e.g., ticks, mites, lice and fleas). These bring emerging or remerging zoonoses closer to humans.

**Table 2. GLM best model selections after 1,050 total computed models based on AICc to determine socio-demographic and environmental factors explaining level of STG, TG and SFG past exposures in the study area.** Similar initial models are applied comprising with the following variables: Landscape type, Gender, Occupation, Age, Educational level, Dog number, Cat number, Pig number, Poultry number and Cattle number (with Poisson distribution after checking data distribution using "fit-distrplus" package in R freeware). Analysis of deviance (type II test) significant level for each variable is indicated as following: *p = 0.05–0.01; **p = 0.01–0.001 and ***p < 0.001. VIF = Variance inflation factor; Wr = Akaike's weights.

| Dependent variable | Explicative variable | Estimate | Standard Error | p-value | VIF | AICc | Wr | Model power | Goodness of fit ($R^2$) |
|---|---|---|---|---|---|---|---|---|---|
| STG titers | Landscape type (upland forested area)*** | 1.212 | 0.013 | < 0.0001 | 1.184 | 9625.156 | 0.109 | 1.000 | 0.999 |
| | Gender (male)*** | 0.778 | 0.010 | < 0.0001 | 1.029 | | | | |
| | Occupation (non-farmer) | -0.019 | 0.012 | 0.107 | 1.045 | | | | |
| | Age*** | 0.035 | 0.001 | < 0.0001 | 1.397 | | | | |
| | Educational level*** | -0.839 | 0.009 | < 0.0001 | 1.412 | | | | |
| | Dog number*** | 0.197 | 0.004 | < 0.0001 | 1.161 | | | | |
| TG titers | Landscape type (upland forested area)*** | -1.771 | 0.031 | < 0.0001 | 1.364 | 7561.437 | 0.048 | 1.000 | 0.989 |
| | Occupation (non-farmer)** | 0.107 | 0.025 | < 0.0001 | 1.022 | | | | |
| | Age*** | 0.023 | 0.001 | < 0.0001 | 1.270 | | | | |
| | Educational level*** | -0.195 | 0.017 | < 0.0001 | 1.286 | | | | |
| | Cattle number*** | 0.087 | 0.004 | < 0.0001 | 1.019 | | | | |
| | Dog number*** | 0.327 | 0.011 | < 0.0001 | 1.311 | | | | |
| SFG titers | Occupation (non-farmer)*** | 0.138 | 0.026 | < 0.0001 | 1.059 | 7640.394 | 0.101 | 1.000 | 0.994 |
| | Age*** | 0.006 | 0.0001 | < 0.0001 | 1.112 | | | | |
| | Educational level*** | -0.312 | 0.016 | < 0.0001 | 1.276 | | | | |
| | Poultry number*** | 0.006 | 0.0001 | < 0.0001 | 1.327 | | | | |

Dogs are one of the prime examples of this phenomenon as they have long been associated with humans for approximately 23,000 years ago, being the first domesticated species on earth [64]. Using a data science approach and network analysis, Morand et al. (2020), [3] revealed that domestic dogs (*Canis lupus familiaris*) were the most important species sharing a great number of potential zoonotic rickettsial species with human (e.g., *R. conorii*, *R. felis*, *R. rickettsii*, *R. typhi*, *O. tsutsugamushi*, *Ehrlichia canis* and *Neorickettsia sennetsu*) when compared with other animals. Recent publications have confirmed the role of domestic dogs as important

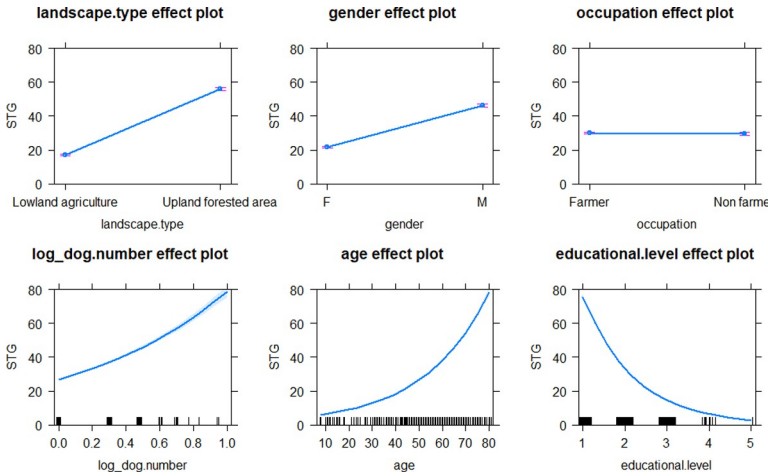

**Fig 3. Effect plots of explicative variables after best model fitting with GLM to estimate level of STG exposures in inhabitants of Saen Thong Subdistrict, Nan Province.** Marks on the X-axis represent individual observations.

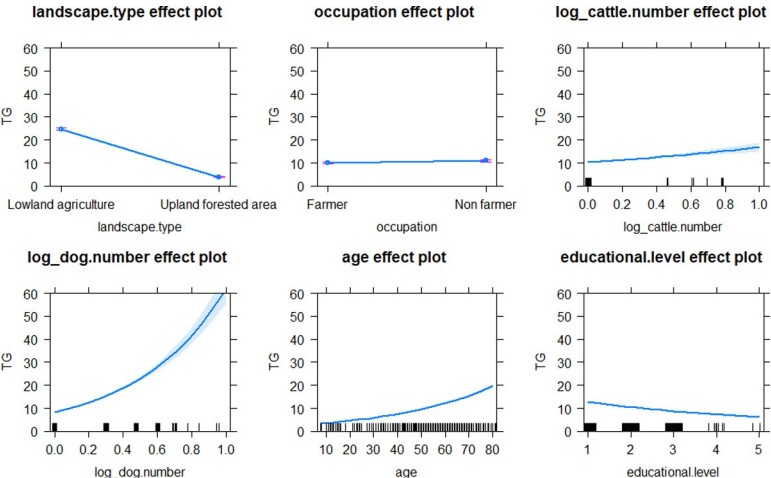

**Fig 4. Effect plots of explicative variables after best model fitting with GLM to estimate level of TG exposures in inhabitants of Saen Thong Subdistrict, Nan Province.** Marks on the X-axis represent individual observations.

reservoirs for *Rickettsia felis* [4,65]. The study, Ng-Nguyen et al. (2020), [4] demonstrated that dogs have ability to sustain *R. felis* for a long period being asymptomatic during the course of infection; and they can also act as biological vehicles passing the rickettsial bacteria to uninfected fleas (*Ctenocephalides felis*) through horizontal transmission. In Sri Lanka, all the three rickettsiosis groups were reported in dogs with a high exposure rates (seroprevalence) at 49% [66]. The information presented here, demonstrates that dogs may play a very important roles in epidemiology and transmission cycle of human rickettsioses. For surveillance and disease ecology aspects, dogs could be used as a sentinel animal to identify high disease burden foci to mitigate risk of rickettsiosis outbreaks in the future.

In addition to dogs, the numbers of cattle and poultry were also positively correlated with levels of human TG and SFG exposures, respectively. Normally, evidence of *R. typhi* exposure in companion pets (e.g., dogs and cats) or in wildlife (e.g., rodents and opossums) translates to increased risk of human infections [22]. To our knowledge, cattle has never been linked

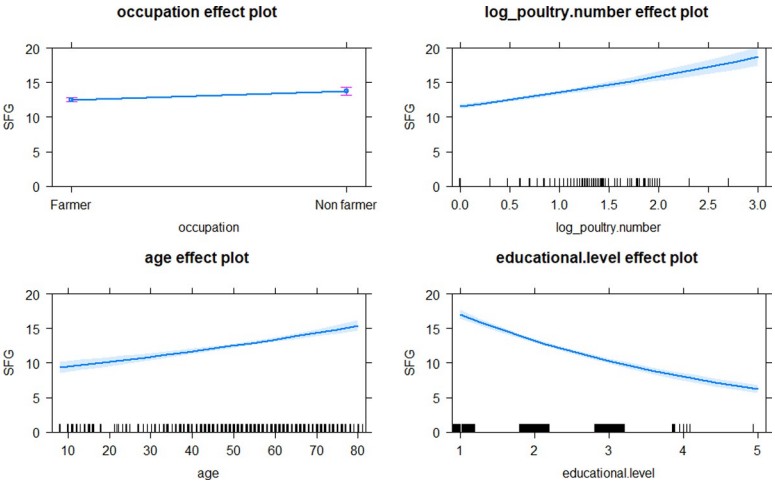

**Fig 5. Effect plots of explicative variables after best model fitting with GLM to estimate level of SFG exposures in inhabitants of Saen Thong Subdistrict, Nan Province.** Marks on the X-axis represent individual observations.

directly with transmission cycle of *R. typhi*. However, previous publications revealed potential occurrences of flea infestation on cattle; e.g., *Ctenocephalides felis felis was found on* buffalo calves in India [67] and on dairy calves in Australia [68]; *Pulex irritans* infested on cow and sheep in Iran [69]. In terms of poultry, several ectoparasite species such as biting lice (e.g., *Lipeurus caponis*, *Goniodes dissimilis*, *G. gallinae*, *Menacanthus stramineus* and *Menopon galli-nae*), fleas (e.g., *Echidnophaga gallinacea*), ticks (e.g., *Haemaphysalis wellingtoni*) and mites (e.g., *Ornithonyssus bursa*, *Dermanyssus gallinae*, *Megninia cubitalis* and *Pterolichus obtusus*) parasitize domestic chickens in Thailand [70,71], and might be play a vectorial role for SFG rickettsial transmission. However, the study of *Rickettsia*-associated poultry is very limited in Southeast Asia. In the other parts of the world, information regarding rickettsial infections either in poultry or in ectoparasites associated with avian species have been documented but only in a limited manner. Antibodies against *Rickettsia* spp. (i.e., *R. rickettsiii*, *R. parkeri* and/ or *R. bellii*) were detected in domestic chickens from Brazil [72]. *Rickettsia sp.* (closely related to *R. africae*) were detected in flea specimens, *E. gallinacea* collected from domestic chickens in Madagascar [73]. SFG rickettsiae (e.g., *R. aeschlimannii*, *R. africae*, *R. helvetica*, *R. massiliae*, *R. monacensis* and R. *slovaca*) were isolated from ticks of migratory birds in European countries [74–76]. Furthermore, *R. helvetica* and *R. monacensis* were detected in ticks removed from birds in Taiwan, alongside with other tick-borne pathogens such as *Borrelia turdi*, *Ana-plasma sp.* and *Ehrlichia sp* [77]. These results highlight a prominent relationship among avian hosts and their ectoparasites in carrying and maintaining SFG rickettsiae in nature.

Further studies on ectoparasite burden in dogs, cattle, poultry and wild rodents as potential reservoirs of several ectoparasites of public health and veterinary importance. These studies should be combined with investigations of rickettsial infections in the animal hosts and their associated arthropod vectors in contrasting ecologies to better determine the role of these animals in zoonotic cycle of rickettsial transmission.

## Supporting information

**S1 Table. Demographic data of each village in Saen Thong Sub-district, Tha Wang Pha District, Nan Province, Thailand.**
(DOCX)

**S1 Fig. Mapping of scrub typhus group (STG) seropositive cases by villages in Saenthong subdistrict, Nan Province.** Green and red circles indicate sero-negativity and sero-positivity of STG, respectively. Size of the red circles indicate level of STG positivity (IFA titers).
(TIF)

**S2 Fig. Mapping of typhus group (TG) seropositive cases by villages in Saenthong subdistrict, Nan Province.** Green and red circles indicate sero-negativity and sero-positivity of TG, respectively. Size of the red circles indicate level of TG positivity (IFA titers).
(TIF)

**S3 Fig. Mapping of spotted fever group (SFG) seropositive cases by villages in Saenthong subdistrict, Nan Province.** Green and red circles indicate sero-negativity and sero-positivity of SFG, respectively. Size of the red circles indicate level of SFG positivity (IFA titers).
(TIF)

## Acknowledgments

We would like to express our gratitude to the local administration in Nan Province, including Nan Provincial Public Health Office, Tha Wang Pha District Public Health Office, Tha Wang

Pha Hospital and Saen Thong Subdistrict Health Promoting Hospital for supporting and facilitating research in the field. Also, we would like to extend our sincere thanks to the community leaders and the villagers for their involvement in this study. CT, AK and SM thank the Thailand International Cooperation Agency (TICA) with the supporting project "Innovative Animal Health".

## Author Contributions

**Conceptualization:** Kittipong Chaisiri, Serge Morand, Stuart D. Blacksell.

**Data curation:** Kittipong Chaisiri, Chuanphot Thinphovong, Malee Tanita, Serge Morand.

**Formal analysis:** Kittipong Chaisiri, Serge Morand, Stuart D. Blacksell.

**Funding acquisition:** Serge Morand, Stuart D. Blacksell.

**Investigation:** Kittipong Chaisiri, Ampai Tanganuchitcharnchai, Anamika Kritiyakan, Chuanphot Thinphovong, Malee Tanita, Serge Morand, Stuart D. Blacksell.

**Methodology:** Kittipong Chaisiri, Ampai Tanganuchitcharnchai, Anamika Kritiyakan, Serge Morand, Stuart D. Blacksell.

**Project administration:** Kittipong Chaisiri, Anamika Kritiyakan, Chuanphot Thinphovong, Malee Tanita.

**Resources:** Kittipong Chaisiri, Ampai Tanganuchitcharnchai, Malee Tanita, Serge Morand, Stuart D. Blacksell.

**Software:** Kittipong Chaisiri, Serge Morand.

**Supervision:** Serge Morand, Stuart D. Blacksell.

**Validation:** Kittipong Chaisiri, Serge Morand, Stuart D. Blacksell.

**Visualization:** Kittipong Chaisiri, Serge Morand, Stuart D. Blacksell.

**Writing – original draft:** Kittipong Chaisiri, Serge Morand, Stuart D. Blacksell.

**Writing – review & editing:** Kittipong Chaisiri, Ampai Tanganuchitcharnchai, Anamika Kritiyakan, Chuanphot Thinphovong, Malee Tanita, Serge Morand, Stuart D. Blacksell.

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
