## [Decision Letter · Decision Letter 0]

8 Nov 2021

Dear Dr. Kittipong Chaisiri,

Thank you very much for submitting your manuscript "Risk factors analysis for neglected human rickettsioses in rural communities in Nan province, Thailand: A community-based observational study along a landscape gradient" for consideration at PLOS Neglected Tropical Diseases. As with all papers reviewed by the journal, your manuscript was reviewed by members of the editorial board and by several independent reviewers. The reviewers appreciated the attention to an important topic. Based on the reviews, we are likely to accept this manuscript for publication, providing that you modify the manuscript according to the review recommendations. 

Sincerely,

Wen-Ping Guo

Associate Editor

Nam-Hyuk Cho

Deputy Editor

Reviewer's Responses to Questions

**Key Review Criteria Required for Acceptance?**

**Methods**

-Are the objectives of the study clearly articulated with a clear testable hypothesis stated?

-Is the study design appropriate to address the stated objectives?

-Is the population clearly described and appropriate for the hypothesis being tested?

-Is the sample size sufficient to ensure adequate power to address the hypothesis being tested?

-Were correct statistical analysis used to support conclusions?

-Are there concerns about ethical or regulatory requirements being met?

Reviewer #1: Methods appear to be fine. I do not have the statistical expertise to critique the methods used, but they seem to be ok at a superficial level.

Reviewer #2: Methodology is well described and objectives are clear. Statistical analysis is correct. No ethical issues

Reviewer #3: Materials and Methods; Pg 7; Line 21 (Blood sample collection)

Please specify the details of the tubes used to collect serum or give the relevant reference

You could change it like this if needed:

Blood samples were collected and processed for rickettsial serology as described previously/elsewhere

**Results**

-Does the analysis presented match the analysis plan?

-Are the results clearly and completely presented?

-Are the figures (Tables, Images) of sufficient quality for clarity?

Reviewer #1: Results are valid

Reviewer #2: The analysis planned and presented matched each other

Results clearly and comprehensively presented

Quality of the images and tables is as per the standard requirements

Reviewer #3: Pg 10; lines 4 & 6, you have mentioned age and occupation could not be assigned in 1.8% and 5.1% respectively. Kindly give a reason/reasons for this. If data was not collected/ or was unclear, you can omit this statement

Pg 10; Line 7-11; Regarding individual interviewing ------ keeping these animals at home, respectively. Sentence is not clear (complicated) please re-write in simple words, so that it is easy to understand.

Pg 10 Line14-15: Simplify this sentence

**Conclusions**

-Are the conclusions supported by the data presented?

-Are the limitations of analysis clearly described?

-Do the authors discuss how these data can be helpful to advance our understanding of the topic under study?

-Is public health relevance addressed?

Reviewer #1: Conclusions appear to be valid and based on the data.

Reviewer #2: Conclusions are supported by the data

Authors have explained the usefulness of the study with reference to Public health

Few limitations could have been added like need for exploring the burden of rodent and ecto parasites in the two contrasting settings

Reviewer #3: (No Response)

**Editorial and Data Presentation Modifications?**

Reviewer #1: The standard of written English is not up to publication level. Suggest review by a native English speaker.

Introduction, line 14. The mites that appear to transmit Orientia sp in Chile are not of the Trombiculidae family.

Material & Methods, line 10/11. What is meant by "no communication problem" ? autism ? poor mobile phone coverage ?

I was surprised that only 6 sera (3.1%) showed cross-reactions between TG-SFG in IF serology. In my lab we see a much higher level of cross-reaction. 

p. 12 line 1/2 "(non-exposure to 4.1%)" What does this mean ?

Table 1. Data is clear but why does village # 8 come after # 3 rather than after # 7 ? It looks untidy.

Reviewer #2: Cut-off titres for ST IgG IFA for epidemiological purposes as opined by few authors may be added.

Reviewer #3: (No Response)

**Summary and General Comments**

Reviewer #1: An interesting study and well performed with clear data.

Reviewer #2: The study is well planned and meticulously carried out

Reviewer #3: Authors have conducted a study which expands the body of knowledge regarding the important factors contributing to rickettsial infections in Asia and Thailand in particular. As the methodology is robust, the results and the conclusions derived are reliable. 

I have a few minor suggestions which are appended as comments.

Another suggestion is that the language and grammar though adequate in most places needs more attention 

Example: Pg12; Line 24 (going to Pg13; Line 1): Lacking of education is recognized as one of poverty indices.

PLOS authors have the option to publish the peer review history of their article (what does this mean?). If published, this will include your full peer review and any attached files.

Reviewer #1: No

Reviewer #2: Yes: SELVARAJ STEPHEN

Reviewer #3: No

Figure Files:

Data Requirements:

Reproducibility:

References

---

## [Decision Letter · Decision Letter 1]

3 Feb 2022

Dear Dr. Chaisiri,

Thank you very much for submitting your manuscript "Risk factors analysis for neglected human rickettsioses in rural communities in Nan province, Thailand: A community-based observational study along a landscape gradient" for consideration at PLOS Neglected Tropical Diseases. As with all papers reviewed by the journal, your manuscript was reviewed by members of the editorial board and by several independent reviewers. The reviewers appreciated the attention to an important topic. Based on the reviews, we are likely to accept this manuscript for publication, providing that you modify the manuscript according to the review recommendations. 

Sincerely,

Wen-Ping Guo

Associate Editor

Nam-Hyuk Cho

Deputy Editor

Reviewer's Responses to Questions

**Key Review Criteria Required for Acceptance?**

**Methods**

-Are the objectives of the study clearly articulated with a clear testable hypothesis stated?

-Is the study design appropriate to address the stated objectives?

-Is the population clearly described and appropriate for the hypothesis being tested?

-Is the sample size sufficient to ensure adequate power to address the hypothesis being tested?

-Were correct statistical analysis used to support conclusions?

-Are there concerns about ethical or regulatory requirements being met?

Reviewer #1: line 9 of p7 states "age over 9 years" but line 12 states "children 7-9 years old". Which statement is correct ? Currently the 2 statements are inconsistent.

line 22 of p7 states "rpm". This is an inappropriate representation of centrifugal force as it can vary with the diameter of the centrifuge rotor. It should always be expressed as " x g ".

Reviewer #2: - Yes the objectives of the study is clearly stated in the testable hypothesis

- Study design is appropriately addressed to the stated objectives

- Sample size is sufficient to ensure adequate power to address the hypothesis being tested

- Statistical analysis used to support conclusions were correct

- Are there concerns about ethical or regulatory requirements being met? No

Reviewer #3: (No Response)

**Results**

-Does the analysis presented match the analysis plan?

-Are the results clearly and completely presented?

-Are the figures (Tables, Images) of sufficient quality for clarity?

Reviewer #1: OK

Reviewer #2: - Does the analysis presented match the analysis plan? - Yes

-Are the results clearly and completely presented? - Yes

-Are the figures (Tables, Images) of sufficient quality for clarity? - Yes

Reviewer #3: (No Response)

**Conclusions**

-Are the conclusions supported by the data presented?

-Are the limitations of analysis clearly described?

-Do the authors discuss how these data can be helpful to advance our understanding of the topic under study?

-Is public health relevance addressed?

Reviewer #1: OK

Reviewer #2: -Are the conclusions supported by the data presented? - Yes

-Are the limitations of analysis clearly described? - Yes

-Do the authors discuss how these data can be helpful to advance our understanding of the topic under study? Yes

-Is public health relevance addressed? - Yes

Reviewer #3: (No Response)

**Editorial and Data Presentation Modifications?**

Reviewer #1: two minor changes only required

Reviewer #2: Accept

Reviewer #3: (No Response)

**Summary and General Comments**

Reviewer #1: OK

Reviewer #2: May be accepted for publication

Reviewer #3: (No Response)

PLOS authors have the option to publish the peer review history of their article (what does this mean?). If published, this will include your full peer review and any attached files.

Reviewer #1: No

Reviewer #2: Yes: SELVARAJ STEPHEN

Reviewer #3: No

Figure Files:

Data Requirements:

Reproducibility:

References

---

## [Editor Report · Decision Letter 2]

12 Feb 2022

Dear Dr. Chaisiri,

We are pleased to inform you that your manuscript 'Risk factors analysis for neglected human rickettsioses in rural communities in Nan province, Thailand: A community-based observational study along a landscape gradient' has been provisionally accepted for publication in PLOS Neglected Tropical Diseases.

Best regards,

Wen-Ping Guo

Associate Editor

Nam-Hyuk Cho

Deputy Editor

---

## [Editor Report · Acceptance letter]

18 Mar 2022

Dear Dr. Chaisiri,

We are delighted to inform you that your manuscript, "Risk factors analysis for neglected human rickettsioses in rural communities in Nan province, Thailand: A community-based observational study along a landscape gradient," has been formally accepted for publication in PLOS Neglected Tropical Diseases.

Best regards,

Shaden Kamhawi

co-Editor-in-Chief

Paul Brindley

co-Editor-in-Chief
